# Silicas with Polyoxyethylene Branches for Modification of Membranes Based on Microporous Block Copolymers

**DOI:** 10.3390/membranes13070642

**Published:** 2023-07-02

**Authors:** Ilsiya M. Davletbaeva, Zulfiya Z. Faizulina, Ekaterina D. Li, Oleg O. Sazonov, Sergey V. Efimov, Vladimir V. Klochkov, Alexander V. Arkhipov, Ruslan S. Davletbaev

**Affiliations:** 1Technology of Synthetic Rubber Department, Kazan National Research Technological University, 68 Karl Marx Str., 420015 Kazan, Russia; faizulina.alin@yandex.ru (Z.Z.F.); katystayls@gmail.com (E.D.L.); sazonov.oleg2010@gmail.com (O.O.S.); 2Institute of Physics, Kazan Federal University, 18 Kremlevskaya Str., 420008 Kazan, Russia; sergej.efimov@kpfu.ru (S.V.E.); vladimir.klochkov@kpfu.ru (V.V.K.); 3Institute of Electronics and Telecommunications, Peter the Great St. Petersburg Polytechnic University, 29 Polytechnicheskaya St., 195251 St. Petersburg, Russia; arkhipov@rphf.spbstu.ru; 4Material Science and Technology of Materials Department, Kazan State Power Engineering University, 51 Krasnoselskaya Str., 420066 Kazan, Russia; darus@rambler.ru

**Keywords:** membranes, block copolymers, modification, polysiloxanes with polyoxyethylene branches, sorption capacity, analytical test systems

## Abstract

We have synthesized cubic and linear polysiloxanes containing polyoxyethylene branches (ASiP-Cu) using tetraethoxysilane, polyoxyethylene glycol, and copper chloride as precursors; the products are stable to self-condensation. The effect of copper chloride content on the chemical structure of ASiP-Cu has been established. A special study was aimed at defining the modifying effect of ASiP-Cu on the sorption characteristics of membranes based on microporous, optically transparent block copolymers (OBCs). These OBCs were produced using 2,4-toluene diisocyanate and block copolymers of ethylene and propylene oxides. The study demonstrated significantly increased sorption capacity of the modified polymers. On the basis of the modified microporous block copolymers and 1-(2-pyridylazo)-2-naphthol (PAN) analytical reagent, an analytical test system has been developed. Additionally, the modified OBCs have the benefit of high diffusion permeability for molecules of organic dyes and metal ions. It has been shown that the volume of voids and structural features of their internal cavities contribute to the complex formation reaction involving PAN and copper chloride.

## 1. Introduction

Membranes have played an important role in analytical chemistry since their discovery in the 1960s, when they began to be widely used as ion-selective electrodes for the determination of cations and anions [1,2]. The membranes for this application are required to ensure fast ion exchange at the interface between the membrane and the solution [3,4,5,6,7,8,9,10,11,12,13,14,15,16]. Block copolymers (BCs) show a considerable promise for the use as sensory membranes. Their properties are largely determined by their morphology which depends on the degree of polymerization of each segment, molecular weights, copolymer polydispersity, and interaction between segments [17,18,19,20,21,22,23,24]. Nanoporous BCs allowing the visual observation of their color change have good prospects of implementation as transparent optical chemical sensors and elements of analytical test systems [25,26]. Diffusion properties of BCs can be controlled by the influence on their supramolecular organization.

Nanoporous, optically transparent block copolymers (OBCs) were synthesized from 2,4-toluylene diisocyanate (TDI) and block copolymers of ethylene and propylene oxides (PPEG). The conditions were determined under which the reaction occurs: the opening of isocyanate groups initiated by the terminal potassium alcoholate groups of PPEG occurs at the thermodynamically more stable carbonyl component. As a result, O-polyisocyanate blocks in coplanar trans-configuration were formed (Figure 1) [27]. 

Further studies have demonstrated that OBCs can be employed as gas-separation membranes, exhibiting high permeability to polar gases and as elements of effective test systems for the qualitative and quantitative determination of heavy metal ions. The processes of microphase separation of OBCs can be enhanced by the addition of silica with polyoxyethylene branches (ASiP-K) as a modifier [28,29,30].

Surface modification of silicon oxide (silica) particles with polymers or oligomers is becoming increasingly popular [31,32]. Polyoxyethylene glycol [33] and polydimethylsiloxane [34] can be used for adsorption interaction with surface -OH groups of silicon dioxide. However, what is most promising is the modification of the surfaces of silicon dioxide nanoparticles with alkoxysilanes [35,36,37,38,39]. Pre-reactions of alkoxysilanes have been used to graft functionalized polyoxyethylene glycols (FPEG) onto silica [40,41] (Figure 2).

Surface-modified silicon dioxide nanoparticles have been used to improve the properties of ultrafiltration membranes. Nanoparticles were obtained by grafting polydimethylsiloxane onto the surface of SiO_2_ nanoparticles using the Steglich reaction. A nanosilicon composite based on SiO_2_ and zirconium dioxide has been modified with polydimethylsiloxane to make the surface superhydrophobic [42,43]. Modified silica particles are mainly obtained by sol–gel synthesis processes, which are traditionally used to produce silica gels, xerogels, and aerogels. This method includes the stage of hydrolysis of alkoxysilanes with the formation of a sol (colloidal system) followed by the condensation of silanol groups and the transformation of the sol into a gel [44,45,46,47,48,49]. However, the application of this method for the manufacture of non-aggregated organosubstituted silica particles (ASiP) presents a hard task because of the high probability of the formation of cross-linked topological structures in the course of the sol–gel processes. The solution to the problem of obtaining ASiPs using alkoxysilanes as the principle reagent consists of the selection of suitable catalysts. As a rule, alkoxysilanes slowly react with water, but the reaction can be accelerated using acidic or basic catalysts [50,51,52,53]. For example, the gelation time for tetraethoxysilane (TEOS) dissolved in ethanol is reduced from 1000 h to 92 via a Cl addition [54]. The use of an alkaline catalyst, on the contrary, noticeably decreases the TEOS hydrolysis rate. The amount of water in the sol–gel system also strongly affects the kinetics of the reactions. At a fixed TEOS concentration, higher water content results in a corresponding increase in the rates of hydrolysis and condensation [55,56,57]. Low water content or high dilution with alcohols can lead to a high yield of oligomers (soluble or volatile) and a reduced yield of SiO_2_ [57]. 

In [57,58] the acid-catalyzed controlled hydrolytic polycondensation of TEOS provided polyethoxysiloxanes with weight-average molecular weights of 2300–11,700, which depend on the reaction molar ratios of the water, catalyst, and solvent to TEOS. They were found to be soluble in common organic solvents and stable to self-condensation, and were characterized with high silica contents of up to 67%.

In [30], organosubstituted silica particles ASiP-K were further employed for OBC modification and were synthesized using TEOS and polyoxyethylene glycol (PEG, MM = 400). The PEG contained 0.1 wt.% of water. ASiP-K synthesis occured via the reactions of TEOS hydrolysis catalyzed by potassium diethylene glycolate (DEG-K), condensation of the resulting Si-OH groups, and subsequent transesterification of the remaining ethoxysilane substituents with polyoxyethylene glycol. The rate and mechanism of the complex reaction was controlled not only by the catalyst’s nature but also by the amount of water returned to the process with each act of condensation of silanol groups. It should be noted that when potassium or sodium hydroxides are used as alkaline catalysts, the rate of the condensation reaction of silanol groups exceeds the rate of the transesterification reaction with the participation of PEG. The use of DEG-K and a very low water concentration softens the reaction conditions and allows the production of individual ASiP-K molecules whose structure represents cubic silica with polyoxyethylene branches.

In connection with earlier results, in this work we used copper chloride (CuCl_2_) as a catalyst and used coordinating center molecules to implement the sol–gel process with the participation of PEG and TEOS. The features of the chemical structure of the synthesized ASiP-Cu have been studied, as well as its modifying effect on the diffusion characteristics of OBC-based membrane materials—the latter was evaluated from the sorption of Rhodamine 6G (R6G) organic dye. A test system was developed employing PAN as the analytical reagent for copper ions.

## 2. Materials and Methods

### 2.1. Materials

The block copolymer of propylene and ethylene oxide (PPEG) was purchased from PJSC Nizhnekamskneftekhim (Nizhnekamsk, Russia). Its formula is HO[CH_2_CH_2_O]_n_[CH_2_(CH_3_)CH_2_O]_m_[CH_2_CH_2_O]_n_K, where n ≈ 4 and m ≈ 48; its molecular weight is 4200 g/mol; it contains 30 wt.% of peripheral polyoxyethylene blocks, where the content of potassium alcoholate groups is 10.9 % from the total number of functional groups and polyoxyethylene glycol (PEG, MM = 400). The PPEG were additionally dried at 85 °C and 0.07 kPa for 2 h down to 0.01 wt.% moisture concentration. 2,4-toluene diisocyanate ≥ 98 wt.% (TDI) was purchased from Sigma-Aldrich (St. Louis, MO, USA). Tetraethoxysilane (TEOS) was purchased from CJSC Vekton (St. Petersburg, Russia). Toluene and CuCl_2_ were obtained from «Component-reaktiv» (Moscow, Russia). Rhodamine-6G (R6G) and 1-(2-pyridylazo)-2-naphthol (PAN) were purchased from Sigma-Aldrich (St. Louis, MO, USA) and used as received. CuCl_2_ was dried at 120 °C in an oven for 120 min until brown.

### 2.2. Synthesis of ASiP-Cu

The reaction was carried out in three stages. In the first stage, 14.46 mL of PEG (water content 0.1 wt.%) and 7.14 mL of TEOS and CuCl_2_ in an amount from 0 to 0.5 wt.% were placed in a round-bottom flask equipped with a stirrer and a thermometer. Synthesis was carried out with stirring at 90 °C for 4 h. At the end of the first stage, stirring was carried out at 90 °C and 15 mmHg for 10 min to remove ethanol. At the second stage, 3.57 mL of TEOS was introduced into the reaction system and stirring was continued at 90 °C for 1 h. At the end of the second stage, stirring was carried out at 90 °C and 15 mmHg for 10 min. At the third stage, 2.1 mL of TEOS was introduced and stirring was continued at 90 °C for 1 h. At the end of the third stage, stirring was also carried out at 90 °C and 15 mmHg for 10 min to remove the resulting ethanol. The yield of products was estimated based on the removal of the calculated amount of ethanol.

### 2.3. Synthesis of PEG-Cu

Synthesis was carried out in a round-bottom flask by mixing PEG with the calculated amount of CuCl_2_ at a temperature of 100 °C until copper chloride was completely dissolved.

### 2.4. Synthesis of OBC

The reaction was carried out at T = 24 °C in a flat-bottomed flask equipped with a back-flow condenser. Stirring was carried out using a magnetic stirrer. PPEG (10 g), toluene (79 mL), bisphenol A (0.04 g), acetic acid (100 μL) and the calculated amount of ASiP-Cu (0–0.5 wt.%) were added to the flask. Then, TDI (4.5 mL) was introduced into the flask. The total amount of anhydrous reagents was 17 wt.%. Ten minutes later, the reaction mass was placed in a Petri dish. The polymer film was cured at room temperature for 72 h.

### 2.5. Preparation of Samples

For immobilization, solutions of R6G and PAN were prepared by dissolving accurately weighed portions in ethyl alcohol with a concentration of R6G 0.005 mol/L and PAN with a concentration of 0.005 mol/L. Ethanol was purchased from MilliporeSigma (St. Louis, MO, USA). R6G was immobilized on OBC via adsorption from R6G alcoholic solutions in a static mode for 10 s. PAN was immobilized on OBC via adsorption from their alcoholic solutions in a static mode with periodic stirring for 4 min. Then, OBC was immobilized with R6G and PAN and they were kept in water for 24 h and dried. 

For the preparation of complexes of CuCl_2_ with PAN immobilized on OBC, CuCl_2_ solution with concentrations of 0.1 g/dm^3^ was prepared by dissolving exact weight of CuCl_2_ in distilled water. The polymer with immobilized PAN was kept in an aqueous solution of CuCl_2_ for 60 min. 

### 2.6. Measurements

#### 2.6.1. Fourier Transform Infrared (FT-IR) Spectroscopy Analysis

The FTIR spectra of the products were recorded on the Nicolet iS20 FT-IR spectrometer (Thermo Fisher Scientific, Waltham, MA, USA) using the attenuated total reflection technique. The spectra were acquired by accumulating 64 scans at a spectral resolution of 4 cm^−1^.

#### 2.6.2. Thermal Gravimetric Analysis (TGA)

TGA was performed using STA-600 TGA–DTA combined thermal analyzer (Perkin Elmer, Waltham, MA, USA). The samples (0.1 g) were loaded in alumina pans and heated at a rate of 5 K/min in a nitrogen atmosphere.

#### 2.6.3. Ultraviolet Visible (UV–Vis) Spectroscopy

Ultraviolet–visible (UV–Vis) spectroscopy was performed with a double-beam spectrophotometer Specord 210 plus (Analytik Jena, Jena, Germany) in quartz cells of 1 and 10 mm length.

#### 2.6.4. Dynamic Light Scattering (DLS)

Dynamic light scattering experiments were carried out on Zetasizer Nano ZS (Malvern, Great Britain). This instrument has a 4 mW He–Ne laser operating at 632.8 nm wavelength. Measurements were carried out at the ambient temperature 25 °C; detection angle was 173°; disposable plastic cuvettes of 1 cm path length were used.

#### 2.6.5. Measurements of the Surface Tension

The droplet-counting method was used to determine the surface tension (σ). The basis of the calculations is the law stating that the mass of a droplet (m) that comes off the pipette of the radius R can be determined as m = 2π·R·σ/g, where g is acceleration of the gravity. This gives us σ = m·g/2π·R for the surface tension coefficient of the tested liquid.

#### 2.6.6. Water Absorption Tests

The method of water absorption is based on the gravimetric determination of the amount of water which is entrapped in an elemental sample. The method of water absorption is based on the gravimetric determination of the amount of water which is entrapped in an elemental sample.

#### 2.6.7. Dynamic Viscosity and Density Measurements

The dynamic viscosity of the samples was determined on SVM 3000 Stabinger Viscometer (Anton Paar, Graz, Austria) with systematic error of ±0.35% of the measured value. At the same time, the density of the samples was determined with systematic error of 0.0005 g cm^−3^.

#### 2.6.8. NMR Spectroscopy

^1^H and ^29^Si NMR spectra of samples were recorded on an Avance II 500 (Bruker) spectrometer. Operating frequency was 99.36 MHz for the ^29^Si channel. An amount of 200 μL of each sample liquid was diluted with 400 μL CDCl_3_ inside standard 5-mm Wilmad WG-1000 NMR tubes. Sample temperature was set to 30 °C. A total of 720 or 800 scans were accumulated in each measurement with the pulse-repetition period of 6.7 s. An additional spectrum of ^29^Si from a sample without any silicon compounds was measured to subtract the residual signal of glass and thus, to improve baseline correction. The total recorded spectral range was from 31.5 to 171.5 ppm.

#### 2.6.9. Water Concentration Measurement

A Mettler Toledo V20 volumetric titrator (Mettler Toledo, Zurich, Switzerland) based on the Karl Fischer method was used to measure water concentration.

#### 2.6.10. Mechanical Loss Tangent Measurements (MLT)

The MLT curves of polymer samples were taken using the dynamic mechanical analyzer DMA 242 (Netzsch, Selb, Germany) operating in the mode of oscillating load. Force and stress–strain correspondence were calibrated using a standard mass. The thickness of the samples was 2 mm. Viscoelastic properties were measured under nitrogen. The samples were heated at the rate of 3 °C/min and frequency of 1 Hz. The mechanical loss tangent was defined as the ratio of the viscosity modulus G″ to the elasticity modulus G′.

#### 2.6.11. Thermomechanical Analysis (TMA)

The thermomechanical curves of polymer samples were obtained using TMA 402 F (Netzsch, Selb, Germany) thermomechanical analyzer in the compression mode. The sample thickness was 2 mm, and the rate of heating was 3 °C/min in the static mode. The load was 2 N.

#### 2.6.12. Contact Angle Measurements

Water contact angle was defined as the angle between the tangent drawn to the surface of the wetting liquid and the wetted surface of the solid. The contact angle of wetting is always measured from the tangent towards the liquid phase. Distilled water was used as the liquid. The membrane was placed on a stand in the measuring cell, which was mounted on a table holder. A drop of water in an amount of 1 mm^3^ was applied to the membrane surface. The droplet height *h* and droplet diameter *d* were determined using an ocular micrometer. The value of cos *θ* was calculated according to:(1)cos θ=d22−h2d22+h2

#### 2.6.13. AFM Studies

Surface topography of samples was imaged on an atomic force microscope (AFM) Nano-DST (Pacific Nanotechnology, Santa Clara, CA, USA) operated in semicontact mode under ambient conditions. Probes NSG01 (by NT-MDT, Russia) were used, with parameters: n-Si, tip curvature 10 nm, frequency ca. 150 kHz, force constant 5.1 N/m.

## 3. Results

### 3.1. Silicas with Polyoxyethylene Branches Characterization

During the synthesis of ASiP-Cu and ASiP-K, the interaction of such thermodynamically incompatible components as TEOS and PEG was visually determined from the moment of disappearance of separation between them. It turned out that the process of ASiP-K formation takes much longer time than formation of ASiP-Cu. For instance, in the presence of CuCl_2_, the separation of TEOS and PEG at 90 °C was broken as soon as 5–7 min after the start of the synthesis. The catalytic effect did not depend on CuCl_2_ content in the range of 0.01–0.5 wt.%. When DEG-K was used as the catalyst at 90 °C, the components completely merged only after three days.

The ASiP-Cu product represents a viscous liquid, soluble in hydrocarbons and emulsifiable in water. It was found that the color of ASiP-Cu changes from light blue to dark brown with an increase in of CuCl_2_ amount in the reaction system based on TEOS and PEG. 

A comparative study of the particle size distribution was carried out for PEG, ASiP-Cu and ASiP-K (Figure 3). The average particle size of PEG in toluene was 450 nm and the particle size distribution itself was narrow. The distribution for ASiP-K was much broader and mainly occupied the range of 60–400 nm with a maximum of 150 nm. However, relatively low-intensity light scattering was observed via the presence of 10–30 nm ASiP-K particles as well as large agglomerates exceeding 1000 nm. 

The content of CuCl_2_ during the synthesis of ASiP-Cu had a significant effect on the average particle size of ASiP-Cu and on the particle size distribution (Figure 3b). At 0.01 wt.% CuCl_2_ content, the intensity of light scattering was approximately 16% near the maximum, corresponding to the particle size of 540 nm. For comparison, for PEG, the light scattering intensity was 23% and the particle size in the region of maximum intensity was 450 nm. The relatively narrow distribution and large particle sizes of polyoxyethylene glycol are explained by the associative interactions typical of PEG. In this regard, the expansion of the particle size distribution accompanied by a decrease in the intensity and an increase in the size of ASiP-Cu particles obtained with a content of 0.01 wt.% CuCl_2_ indicates not only the formation of large particles, but also the implementation of associative interactions involving polyoxyethylene branches in the ASiP–CuCl_2_ structure.

A further increase in the content of CuCl_2_ during the preparation of ASiP-Cu led to a decrease in the mean size of the resulting particles, which reaching the minimum value of 200 nm at 0.5 wt.% CuCl_2_ (Figure 3b). The most likely reason for the ASiP-Cu particles size reduction is the coordination binding of polyoxyethylene branches and the subsequent decrease in the size of agglomerates.

Kinematic viscosity (η) of PEG was 120 cSt, while for ASiP-Cu obtained with a CuCl_2_ content of 0.01 wt.%, the η value grew to 300 cSt. A further increase in the content of CuCl_2_ led to ASiP-Cu kinematic viscosity reduction in the range from 100 to 70 cSt. Thus, the rheological test results correlated with particle size measurements. 

The ASiP-Cu final color depended on the content of copper chloride introduced into the reaction system. At low concentrations (up to 0.08 wt.%), ASiP-Cu was blue. With an increase in the proportion to 0.1–0.15 wt.%, the ASiP-Cu changed to dark green, and with further increases in CuCl_2_ (from 0.3 wt.%), the color of the final product became brown. 

The extinction coefficient in the electronic spectra noticeably increased with increasing wavelength. Therefore, its measurements for ultraviolet and visible regions were carried out with cuvettes of different thickness, and Figure 4 presents two series of such spectra for the same products.

According to electron spectroscopy, at a content of 0.01 wt.% CuCl_2_ in the composition of ASiP-Cu, the spectrum of this compound appears only in the ultraviolet region and almost coincides with the spectrum of polyoxyethylene glycol. The fact that in the UV region, the spectrum of ASiP-Cu obtained at a content of [CuCl_2_] = 0.01–0.03 wt.% practically coincides with the spectrum of polyoxyethylene glycol indicates the absence of complex formation in ASiP-Cu-(0.01–0.03). That is, ASiP-Cu obtained at [CuCl_2_] = 0.01–0.03 wt.% exists in the form of polysiloxanes with polyoxyethylene branches, in which polyoxyethylene glycol branches do not enter into coordination binding with Cu (II) ions, and CuCl_2_, upon interaction of TEOS with PEG, performs only the function of a catalyst. At the same CuCl_2_ content, an almost three-fold increase in kinematic viscosity of ASiP-Cu-0.01 was observed, as well as an increase in particle size (Figure 3).

When the content of [CuCl_2_] = 0.1 wt.%, an abrupt change in the shape of the spectrum occurs, accompanied by the appearance of a new absorption band in the region of 270–280 nm. With an increase in the content of CuCl_2_ from 0.1 to 0.5 wt.% during the synthesis of ASiP-Cu, the intensity of the absorption band in the region of 270–280 nm grows (Figure 4a). The fact that, during the formation of ASiP-Cu-(0.1–0.5), a complexation reaction occurs is evidenced by the appearance of new absorption bands at 500 and 900 nm, which increase in intensity with an increase in the CuCl_2_ content in ASiP-Cu-(0.1–0.5) (Figure 4b).

Analysis of the results of measuring the size of ASiP-Cu particles, kinematic viscosity and electronic spectra allows us to conclude that CuCl_2_ at 0.01 wt.% catalyzes the sol–gel process, leading to the formation of ASiP-Cu, but at such a low content, copper chloride does not enter into complex interactions with PEG, which is an integral part of the ASiP-Cu structure. An increase in the content of copper chloride to 0.1 wt.% involves a significant change in the structural organization of ASiP-Cu. In this case, copper chloride coordinates with open-chain analogues of crown ethers by trapping Cu(II) ions into the crown cavity created by polyoxyethylene glycol branches of ASiP-Cu particles [59,60].

To explain the results obtained, the products of the interaction of CuCl_2_ with PEG (PEG-Cu) were studied. According to the electronic spectra for PEG-Cu-(0.01–0.08) shown in Figure 5, there are practically no changes in the UV region of the spectrum in comparison with PEG. A band appears in the visible region at 700 nm, which is characteristic of copper chloride aquacomplexes. 

Starting from PEG-Cu-0.1, the spectra of PEG-Cu-(0.1–0.5) (Figure 5) become similar to the electronic spectra of ASiP-Cu-(0.1–0.5) (Figure 4). Thus, at a content of [CuCl_2_] = 0.1 wt.% for the PEG-Cu-(0.1–0.5), similarly to ASiP-Cu-(0.1–0.5), a new absorption band appears in the region of 270–280 nm (Figure 5a). It is known [61] that the appearance of a spectrum in this region of the ultraviolet range 270–280 nm can be a consequence of n→π * transitions due to the structure of organic molecules. Since the organic component in PEG-Cu is polyoxyethylene glycol, the observed changes in the spectrum may result from the involvement of polyoxyethylene glycol in the coordination binding. In addition, the evidence for the occurrence of complex formation for the PEG-Cu-(0.1–0.5) system is the appearance and increase in the intensity of the absorption band in the region of 900 nm (Figure 5b). Thus, the PEG–Cu-(0.1–0.5) system, similarly to ASiP-Cu-(0.1–0.5), is characterized by the processes of coordination binding of polyoxyethylene glycol by Cu(II) ions.

For PEG-Cu in the range of content in the system [CuCl_2_] = 0.01–0.5, the particle sizes (Figure 6), viscosity and density (Figure 7) were measured. For the PEG-Cu-(0.01–0.08) system, all these parameters grew with CuCl_2_ content, which can be explained by the occurrence of associative intermolecular interactions between CuCl_2_ and PEG.

Further increases in particle size, viscosity, and density with the growing content of CuCl_2_ for the interaction products in the PEG–Cu-(0.1–0.5) system, as well as enhanced absorption intensity in the electronic spectra (Figure 5), can be ascribed to the involvement of chloride bridges in the formation of coordination structures.

In the introduction, we mentioned a previous study [37], wherein ASiP-K was obtained on the basis of TEOS, PEG, and DEG-K. The structure of AsiP-K includes cubic silica with polyoxyethylene branches. It has been shown that the ^29^Si NMR spectrum of AsiP-K contains a resonance from ^29^Si nuclei with the chemical shift of –82.0 ppm, corresponding to silicon atoms in the composition of TEOS. Low signal intensity in the region of δ = –82 ppm in ^29^Si NMR spectra of AsiP-K confirms the practical absence of TEOS in the composition of AsiP-K. The appearance of a signal at δ = –108 ppm correlates with the formation of a cubic silica derivative. 

In this work, ^1^H NMR and ^29^Si NMR spectra were measured for TEOS, AsiP-Cu-0.01, AsiP-Cu-0.1, and ASiP-Cu-0.5 (Figure 8). The ^29^Si NMR spectrum of ASiP-Cu-0.01 contains signals with δ(^29^Si) = –82.3 and –89.2 ppm; also, a low-intensity signal with δ(^29^Si) = –96.6 ppm can be seen. The broad background peak around –108 ppm with the width at half-maximum of ~30 ppm is associated with absorption in the NMR tube glass [62]. Relative intensity of resonance signals from ^29^Si nuclei at –82.3 and –89.2 ppm for sample 3 (ASiP-Cu-0.5) is increased (the background signal looks weaker at the same number of scans), and in sample 4 (ASiP-Cu-0.5), these two signals achieve their maximum intensity. There is also a weak and relatively broad resonance line at –96.4 ppm, which is hardly seen in sample 2 but is noticeable in samples 3 and 4. Studies [57,58] describe polyethoxysiloxanes as consisting of siloxane backbones with side chains of ethoxy and silanol groups obtained using the acid-catalyzed controlled hydrolytic polycondensation of TEOS. According to the ^29^Si NMR spectra, the signals at –89.2 ppm and –92.4 ppm was associated with polyethoxysiloxane consisting of siloxane backbones with side chains of ethoxy and silanol groups.

For sample 2 (ASiP-Cu-0.01), the signal in the region of −108 ppm exceeds the background intensity. The peak is noticeably expanded and shifted to δ(^29^Si) = –112 ppm. It should be noted that larger nanoparticles representing ASiP-Cu agglomerates would be too heavy for ^29^Si NMR, so the corresponding signals might be broadened to complete disappearance. Therefore, the ^29^Si NMR spectroscopy measurements performed in this study allowed us to discern the formation of only a linear polysiloxane core from the presence of signals from ^29^Si nuclei at –89.2 and –96.6 ppm. To confirm the possibility of the existence of a polysiloxane core in the cubic topology in the composition of ASiP-Cu, additional measurements of surface tension and TGA were carried out. 

The ^1^H NMR spectrum of TEOS includes peaks of the methyl group at δ(^1^H) = 1.1 ppm and of the methylene group at δ(^1^H) = 3.7 ppm. Judging by the ^1^H NMR and ^29^Si NMR spectra, the ASiP-Cu-0.01, ASiP-Cu-0.1 and ASiP-Cu-0.5 samples contained some amount of unreacted TEOS. However, the observed upward shift of the peaks of the methyl group protons in ASiP-Cu-0.01, ASiP-Cu-0.1, and ASiP-Cu-0.5 is a consequence of an enhancement of their shielding.

The results of ^1^H NMR spectroscopy correlate with the data of FT-IR spectroscopy. Figure 9a shows fragments of the FT-IR spectra in the region of deformational vibrations of methyl and methylene groups, and Figure 9b shows the region of their stretching vibrations. For TEOS, the –CH_3_ stretching vibrations produce the peak at 2990 cm^−1^, and the –CH_2_ stretching vibrations correspond to the peak at 2890 cm^−1^. The deformational vibrations of the –CH_3_ and –CH_2_ groups appear at 1395 and 1367 cm^−1^, respectively. The identification of absorption bands corresponding to the stretching and bending vibrations of –CH_3_ and –CH_2_ groups in the FT-IR spectra was carried out using [63]. As a result of the interaction of TEOS and PEG, the intensity of the bands at 2990 cm^−1^ and 1395 cm^−1^ wass noticeably reduced, but the bands themselves did not completely disappear. The decrease in the intensity of the bands at 2990 cm^−1^ and 1395 cm^−1^ is a consequence of the fact that the main part of ethanol was removed from the products of the interaction of TEOS and PEG.

The analysis of the FT-IR and ^29^Si NMR spectra allows us to describe the structural features of ASiP-Cu-0.01, ASiP-Cu-0.1, and ASiP-Cu-0.5 as follows. At [CuCl_2_] = 0.01 wt.%, copper chloride only performs the function of a catalyst and does not enter into coordination interaction with PEG. As the CuCl_2_ content grows, the possibility of forming a silica core in the cubic form (structure I) decreases, and the polysiloxane core becomes linear (structure II). An increase in the content of CuCl_2_ to 0.5 wt.% leads to the coordination binding of polyoxyethylene glycol by Cu(II) ions (structure III) (Figure 10).

Figure 11 presents the results of thermogravimetric analysis. Comparison of the TGA curves for PEG, ASiP-K, and ASiP-Cu leads to the conclusion that the main part of PEG was incorporated into ASiP-K and ASiP-Cu. The similarity of the chemical structure of ASiP-K and ASiP-Cu-0.01, which are characterized by the existence of a silica core in a cubic form, is confirmed by the almost coinciding of the TGA curves for these compounds. The thermal resistance of ASiP-Cu-0.1 and ASiP-Cu-0.5, predominantly consisting of coordinated linear polysiloxanes with polyoxyethylene branches, turned out to be higher than the heat resistance of ASiP-K and ASiP-Cu-0.01 due to significant differences in their chemical structure. Based on the calculation of the material balance and the boiling points of ethanol (78 °C) and TEOS (168 °C), the residual content of EtOH and TEOS in the studied compounds has been estimated. For ASiP-Cu-0.5, the loss of EtOH turned out to be the lowest in the series under study and amounted to 4 wt.%, the figure of TEOS loss was 11 wt.%. For ASiP-Cu-0.01, the loss of EtOH was 7 wt.%, and loss of TEOS—13 wt.%. For ASiP-K, the loss of EtOH turned out to be the highest and amounted to 10 wt.%, with 17 wt.% for TEOS. Thus, we can conclude that TEOS achieves the highest conversion during the synthesis of ASiP-Cu-0.5.

Similarly to ASiP-K, the ASiP-Cu samples obtained in this work also exhibited amphiphilic character. Surface active properties of ASiP-Cu compounds depended on the features of their chemical structure controlled by the amount of CuCl_2_ used in the synthesis. Cubic and linear polysiloxanes containing polyoxyethylene branches exhibited higher amphiphilicity compared to their coordinated linear counterparts. The lowest value of surface tension and critical concentration of agglomerate formation (Figure 12) was observed for ASiP-Cu-0.01. Further, these values increased for ASiP-Cu-0.1 and ASiP-Cu-0.5.

### 3.2. Polymer Sorption Capacity

Sorption characteristics for OBC and for modified OBC were studied, for this purpose the dye Rhodamine 6G (R6G) was immobilized on film polymer samples. Figure 13 shows spectra of R6G in two states: in an alcoholic solution and when immobilized on the OBC. The features of the supramolecular organization of OBC determine the chemical structure of the inner cavity of microsized voids and produce significant changes in the spectrum of the sorbed dye. For instance, the spectral maximum of R6G in an alcoholic solution is in the region of 526 nm (Figure 13 line 1), while immobilization shifts it to 538 nm (Figure 13 line 2). Similar phenomena are important when using dyes for the quantitative analysis of heavy metal ions. To enhance the analytical effect, surfactants are currently used which contribute to an increase in the intensity of the spectrum and the shift of the spectral range of the organic color indicator necessary for analysis. This phenomenon is associated with the formation of intermolecular bonds of the dye with the hydrophobic component of surfactants. In the case of OBC, the effect of using surfactants is achieved by the peculiarities of the chemical structure of the inner surface of the pores.

The use of ASiP-Cu modifiers leads to a noticeable increase in the sorption capacity of OBC depending on the structure of ASiP-Cu and its amount in the OBC. The most significant intensity increase near the maximum in the electronic spectrum of immobilized R6G was observed when we used ASiP-Cu-0.1 or ASiP-Cu-0.5 (Figure 13b,c). For ASiP-K (Figure 13d), the intensity of the spectrum of the modified OBC practically did not change; however, a slight bathochromic shift was observed.

The growth of the R6G dye absorption intensity correlates with the decrease in the contact angle of polymers observed when the content of modifiers in their composition was increased (Figure 14a). Modification of block copolymers using ASiP-Cu enhanced their water absorption capability(Figure 14b), indicating an increase in the free volume of OBC.

The use of OBC modified with 0.5 wt.% ASiP-Cu-0.5 allowed us to obtain a test system, immobilizing the PAN analytical reagent on the polymer (Figure 15). It was found that PAN molecules quickly fill the free space of the polymer; the required level of the sorption coefficient was reached within 4 min after the OBC sample was immersed in 0.005 mol/L PAN solution. Further treatment of the polymer leads to its rapid saturation with the analytical reagent, which reflects high diffusion characteristics of the modified OBC.

Before performing the analytical reaction for the determination of CuCl_2_, the polymer with immobilized PAN was preliminarily kept for one day in water. This procedure was necessary to prevent the analytical reagent desorption and its reaction with the metal ions to be determined directly in the solution. The appearance of a new band at 570 nm for the test system kept in a CuCl_2_ solution indicates that microvoids formed in the OBC volume can capture not only dye molecules, but the detectable metal ions as well. In addition, the volume of voids and structural features of their internal cavity contribute to the complexation reaction involving PAN and CuCl_2_. The most significant changes were observed for OBC modified with ASiP-Cu-0.5.

For OBC and modified OBC, thermomechanical and dynamic mechanical curves were obtained (Figure 16). Along with ASiP-Cu, we have also used ASiP-K as a modifier. The content of modifiers was 0.5 wt.% for all samples under study, which was justified by the highest sorption capacity demonstrated by OBC for this modifier amount. We observed significant effect of modification on TMA and DMA curves reflecting changes in the supramolecular organization of OBC.

As shown above, for ASiP-Cu-0.01 and ASiP-K the polysiloxane core has cubic topology (Figure 10, structure I). For ASiP-Cu-0.1 and ASiP-Cu-0.5, polysiloxane is linear (Figure 10, structure II), and a part of the polyoxyethylene branches are coordinated by copper ions (Figure 10, structure III).

It was shown that the TGA curves (Figure 11) for OBC obtained using ASiP-Cu-0.01 and ASiP-K practically coincide. It turned out that the TMA and DMA curves also show similarity. The presence of two regions of relaxation transitions on the TMA curves indicates the hierarchical nature of the formation of the supramolecular structure of the modified OBC.

For OBC obtained using ASiP-Cu-0.1, relaxation transitions in the region of 220–250 °C became less pronounced. This may be explained by the increased rigidity of the framework built by O-polyisocyanate blocks. The largest increment of coplanar O-polyisocyanate framework rigidity occurred when ASiP-Cu-0.5 was used.

The most likely reason for the observed effect of ASiP-Cu on the supramolecular organization of OBC is the cooperative effects arising from the associative interaction of modifiers with the peripheral polyoxyethylene component of the OBC flexible block, which is directly associated with rigid O-polyisocyanate blocks.

The surface morphology of the samples was studied using atomic force microscopy (Figure 17). AFM images showed the porous structure of OBC with relatively deep depressions on the surface of the samples. Modification of OBC using ASiP-Cu led to noticeable changes in surface morphology for the modified OBC depending on the structure of ASiP-Cu. The use of the ASiP-K modifier caused reduction of the number and size of surface depressions on OBS samples.

## 4. Conclusions

Using tetraethoxysilane, polyoxyethylene glycol and copper chloride, silicas with polyoxyethylene branches stable to self-condensation were synthesized. It was established that copper(II) chloride at its low content (0.01 wt.%) catalyzes the formation of ASiP with a cubic silica core. With an increase in the content of CuCl_2_ to 0.1 and 0.5 wt.%, the silicas in the composition of ASiP are linear, and copper (II) ions enter into a complex-forming interaction with the polyoxyethylene branches of ASiP.

The modifying effect of ASiP on the sorption characteristics of membranes based on microporous block copolymers, obtained using 2,4-toluene diisocyanate and block copolymers of ethylene and propylene oxides, were studied. It was established that the greatest effect on the supramolecular organization, surface morphology, and increase in the sorption capacity of OBC was exerted by ASiP-Cu-0.5.

A test system was obtained using the modified OBC and the PAN analytical reagent. The high diffusion permeability of modified OBC for molecules of organic dyes and metal ions was established. It was shown that the volume of voids and structural features of their internal cavity contribute to the complexation reaction involving PAN and CuCl_2_. The best results were obtained for OBC modified with ASiP-Cu-0.5. 

## Figures and Tables

**Figure 1 membranes-13-00642-f001:**
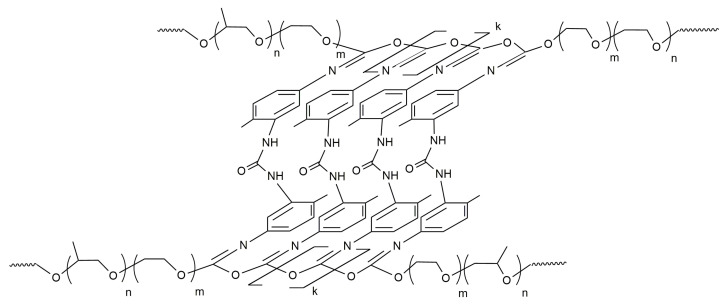
Structure of OBC synthesized based on 2,4-toluylene diisocyanate and BCs of ethylene and propylene oxides.

**Figure 2 membranes-13-00642-f002:**
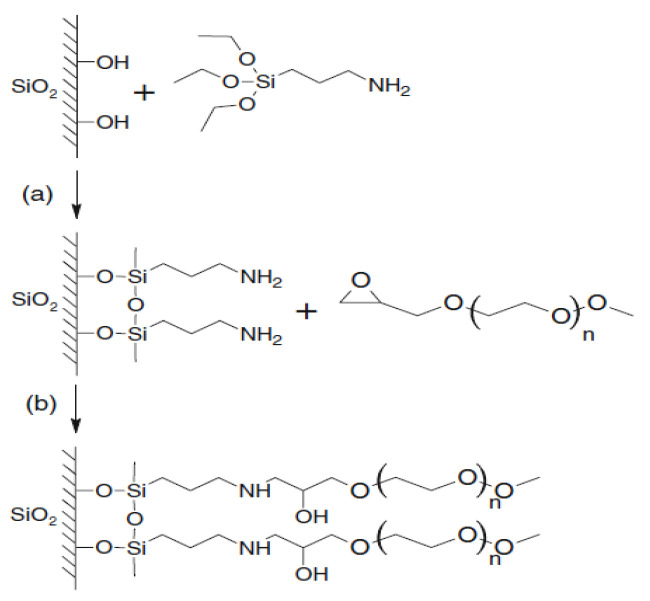
Scheme of aminopropyltriethoxysilane (AGM) and FPEG grafting onto a quartz surface: (**a**) the interaction reaction of the quartz surface with AGM; (**b**) the interaction reaction a substrate containing aminopropylethoxysilane branches with FPEG.

**Figure 3 membranes-13-00642-f003:**
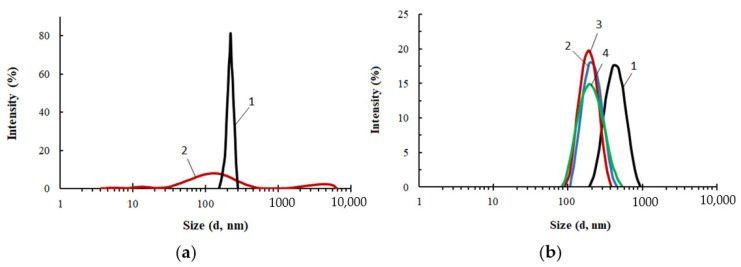
Particle size distribution for PEG (1), ASiP-DEG-K (2)—(**a**); for ASiP-Cu, obtained at the content of CuCl_2_: 0.01 (1), 0.1 (2), 0.3 (3), 0.5 (4) wt.%—(**b**). Toluene, C_ASiP-Cu_ = 1 wt.%, T = 20 °C.

**Figure 4 membranes-13-00642-f004:**
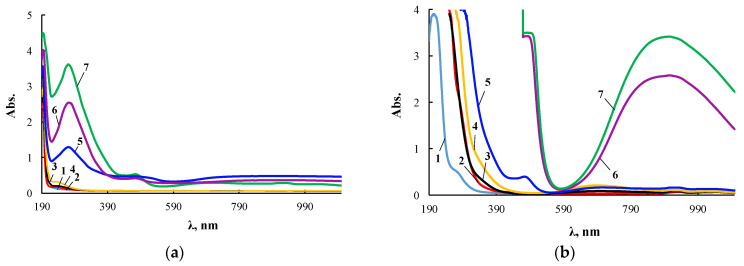
Electronic spectra of PEG (1), ASiP-K (2), and ASiP-Cu obtained at [CuCl_2_] = 0.01 (3), 0.03 (4), 0.1 (5), 0.3 (6), 0.5 (7) wt.% applied in thin layer (**a**) and of PEG (1), ASiP-K (2), and ASiP-Cu obtained at [CuCl_2_] = 0.01 (3), 0.03 (4), 0.1 (5), 0.3 (6), 0.5 (7) wt.% applied in thick layer (**b**).

**Figure 5 membranes-13-00642-f005:**
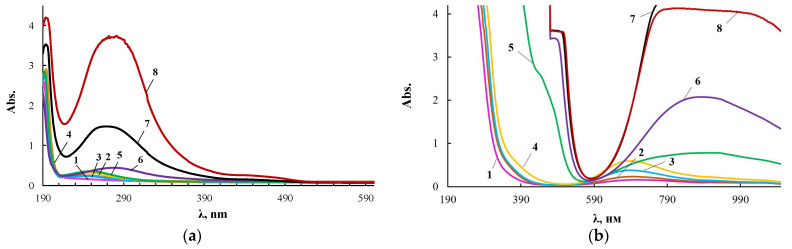
Electronic spectra of the PEG-Cu obtained with [CuCl_2_] = 0.01 (1), 0.03 (2), 0.05 (3), 0.08 (4), 0.1 (5), 0.3 (6), 0.5 (7), 1.0 (8) wt.% applied in thin layer (**a**) and of the PEG-Cu obtained with [CuCl_2_] = 0.01 (1), 0.03 (2), 0.05 (3), 0.08 (4), 0.1 (5), 0.3 (6), 0.5 (7), 1.0 (8) wt.% applied in thick layer (**b**).

**Figure 6 membranes-13-00642-f006:**
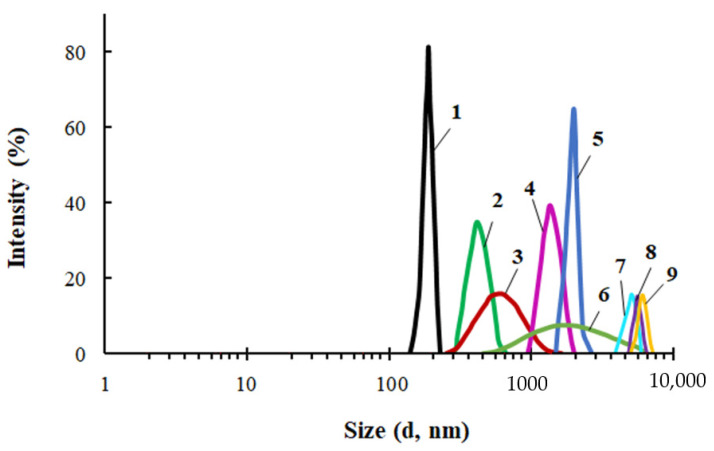
Particle size distribution for PEG (1) and PEG-Cu obtained at [CuCl_2_] = 0.01 (2), 0.03 (3), 0.05 (4), 0.1 (5), 0.2 (6), 0.3 (7), 0.4 (8), 0.5 (9) wt.%. Toluene, C_ASiP-Cu_=0.5 wt.%, T = 20 °C.

**Figure 7 membranes-13-00642-f007:**
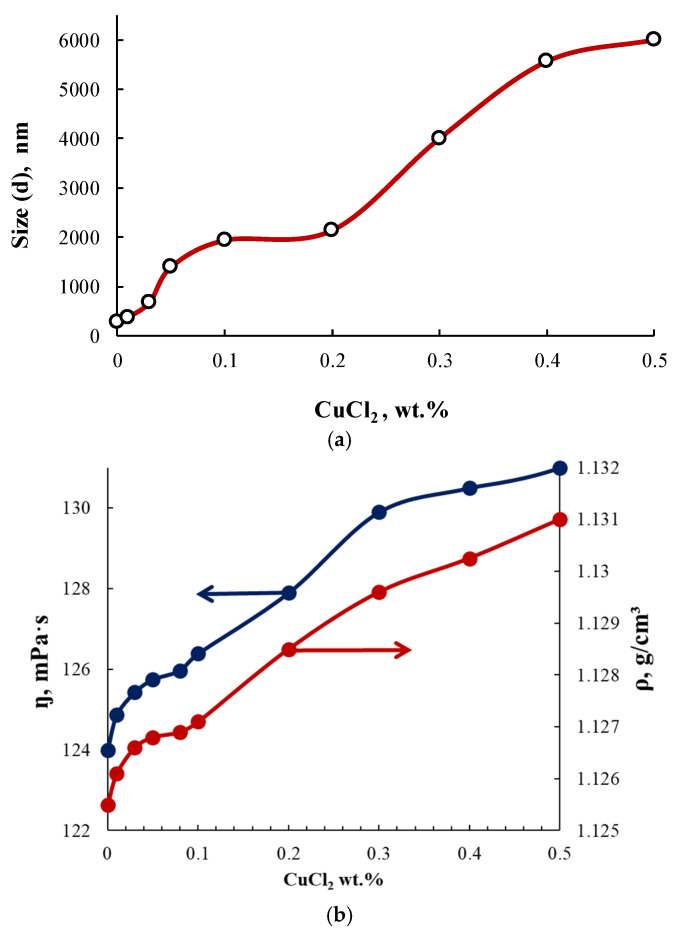
Dependences of the size of PEG-Cu particles (**a**), dynamic viscosity, and density of PEG-Cu (**b**) on the content of CuCl_2_ used in the PEG-Cu synthesis.

**Figure 8 membranes-13-00642-f008:**
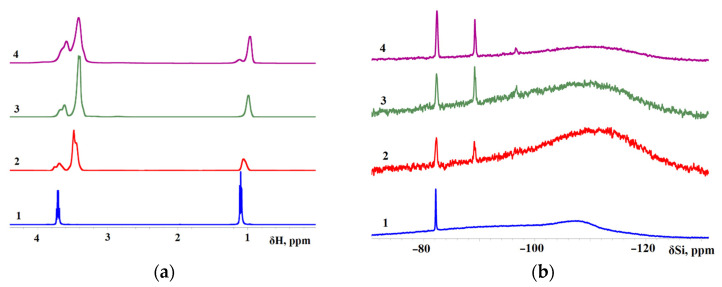
^1^H NMR (**a**) and ^29^Si NMR (**b**) spectra for TEOS (1), ASiP-Cu-0.01 (2), ASiP-Cu-0.1 (3), ASiP-Cu-0.5 (4). ^1^H frequency is 500 MHz; T = 30 °C.

**Figure 9 membranes-13-00642-f009:**
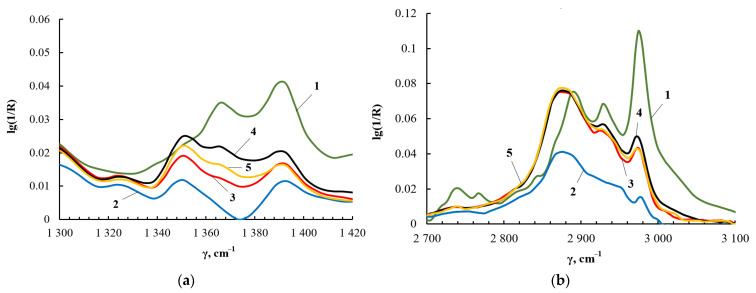
Fragments of the FT-IR spectrum of TEOS (1), ASiP-K (2), ASiP-Cu-0.01 (3), ASiP-Cu-0.1 (4), ASiP-Cu-0.5 (5) in the range from 1300 to 1420 cm^−1^ (**a**) and of TEOS (1), ASiP-K (2), ASiP-Cu-0.01 (3), ASiP-Cu-0.1 (4), ASiP-Cu-0.5 (5) in the range from 2700 to 3100 cm^−1^ (**b**).

**Figure 10 membranes-13-00642-f010:**
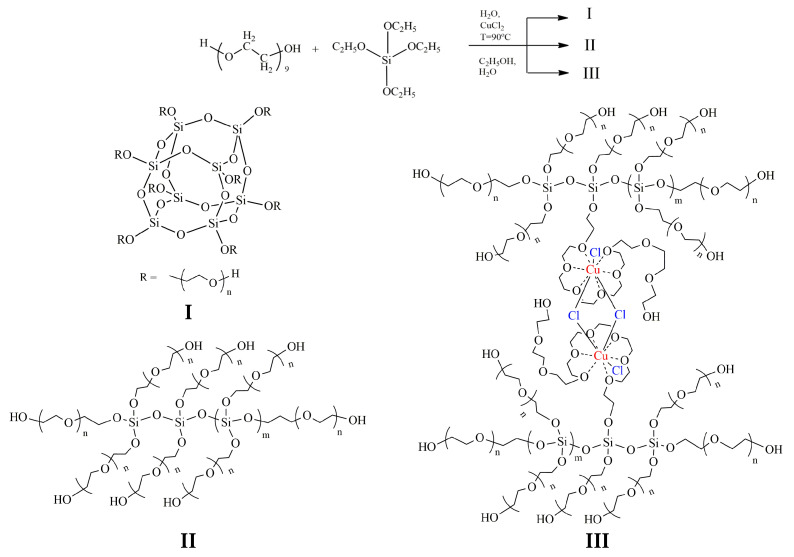
Scheme of polysiloxanes with polyoxyethylene branches formation. ASiP-Cu structure of silica cubic topology (**I**), ASiP-Cu structure of silica linear topology (**II**), ASiP-Cu structure of silica linear topology with polyoxyethylene glycol branches coordinated by copper ions (**III**).

**Figure 11 membranes-13-00642-f011:**
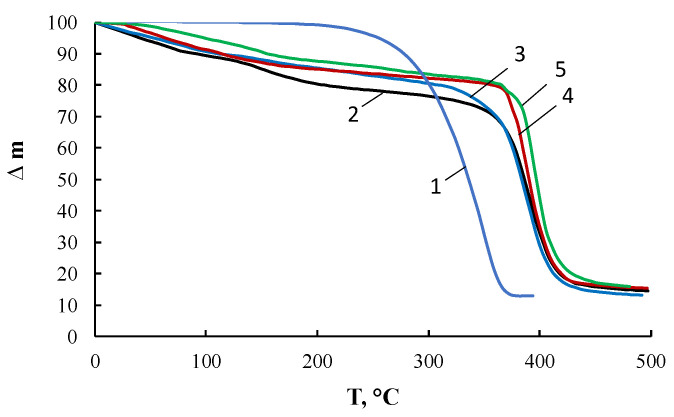
TGA curves for PEG (1), ASiP-K (2), ASiP-Cu-0.01 (3), ASiP-Cu-0.1 (4), ASiP-Cu-0.5 (5).

**Figure 12 membranes-13-00642-f012:**
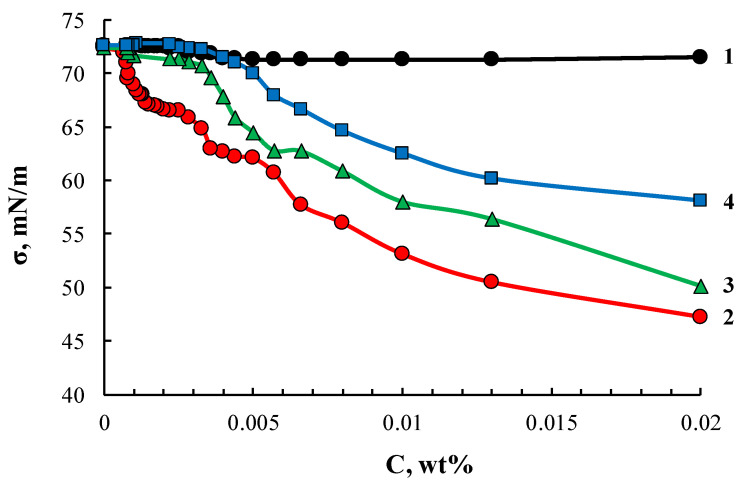
Surface tension isotherms for PEG (1), ASiP-Cu-0.01 (2), ASiP-Cu-0.1 (3), ASiP-Cu-0.5 (4).

**Figure 13 membranes-13-00642-f013:**
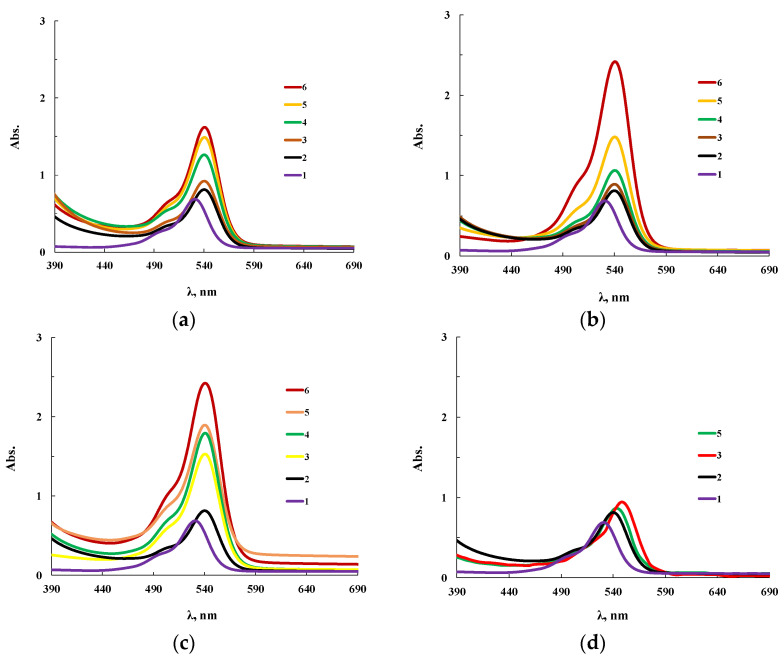
Electronic absorption spectra of an alcoholic solution of R6G (0.005 mol/l) (1) and R6G immobilized on OBC modified with ASiP-Cu-0.5 (**a**), ASiP-Cu-0.1 (**b**), ASiP-Cu-0.01(**c**) and ASiP-K (**d**). The content of modifiers in the polymer are 0.00 (2), 0.05 (3), 0.15 (4), 0.30 (5), 0.50 (6) wt.%.

**Figure 14 membranes-13-00642-f014:**
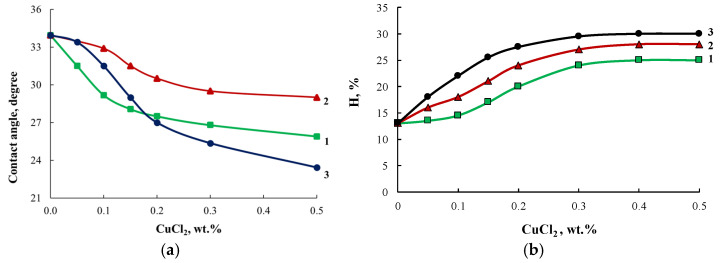
Contact angle (**a**) and water absorption (H, %) (**b**) curves for OBC modified with ASiP-Cu-0.01 (1), ASiP-Cu-0.1 (2), ASiP-Cu-0.5 (3) depending on the ASiP-Cu-(0.1–0.5) content.

**Figure 15 membranes-13-00642-f015:**
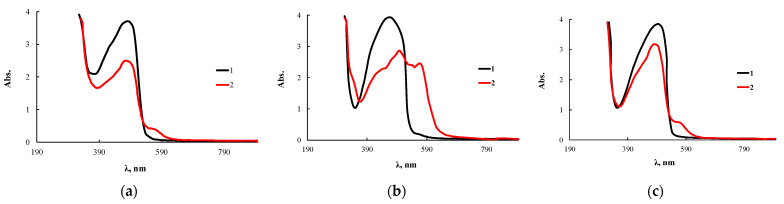
Electronic absorption spectra of PAN (1) and complexes of PAN with CuCl_2_ (2) immobilized on OBC (**a**) and OBC, obtained using 0.5 wt.% of ASiP-Cu-0.5 (**b**) and ASiP-K (**c**).

**Figure 16 membranes-13-00642-f016:**
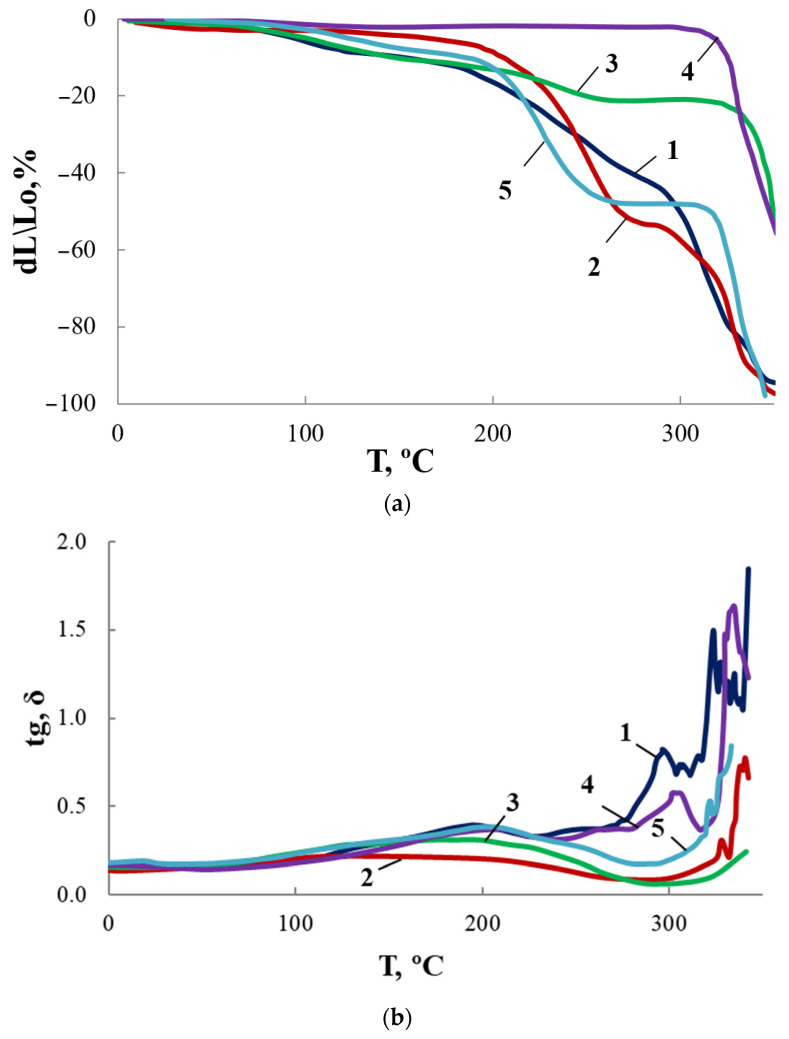
TMA (**a**) and MLT (**b**) curves for OBC (1) and OBC modified with 0.5 wt.% of ASiP-Cu-0.01 (2), ASiP-Cu-0.1 (3), ASiP-Cu-0.5 (4), ASiP-K (5).

**Figure 17 membranes-13-00642-f017:**
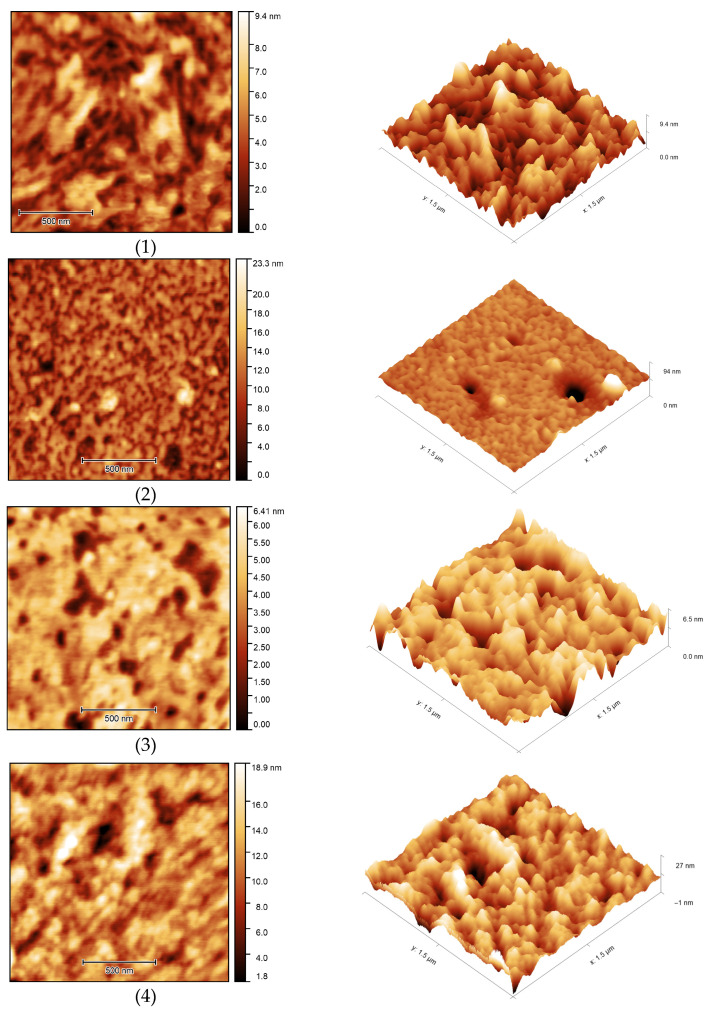
AFM images of OBC (**1**) and OBC modified with 0.5 wt.% of ASiP-Cu-0.01 (**2**), ASiP-Cu-0.1 (**3**), ASiP-Cu-0.5 (**4**), ASiP-K (**5**).

## Data Availability

The data presented in this study are available on request from the corresponding author.

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
