# Peer review of "Silicas with Polyoxyethylene Branches for Modification of Membranes Based on Microporous Block Copolymers"

_membranes, 2023, doi:10.3390/membranes13070642_

Round 1

Reviewer 1 Report

Firstly, the lack of a background abstract makes it difficult to understand your research. Secondly, some molecular structures and data diagrams should be  provided with clear versions. What's more, the ordinate of UV Electronic spectra(such as figure 4,5,13 and 15) should be clear and precise. Finally, the study lacked innovation and the presentation of the article lacked logic.

The English language needs to be revised for clarity.

Author Response

Thank you for your comments and feedback, which have helped us to substantially improve our manuscript. We carefully investigated you recommendations and prepared comments for each item you mentioned in your review.

Firstly, the lack of a background abstract makes it difficult to understand your research. Secondly, some molecular structures and data diagrams should be provided with clear versions. What's more, the ordinate of UV Electronic spectra (such as figure 4,5,13 and 15) should be clear and precise. Finally, the study lacked innovation and the presentation of the article lacked logic.

Answer:   Additional background information has been added to the text. Figure 1 has been re-arranged. In Figures 4,5,13 and 15, the ordinate has been changed in accordance with the comments. The abstract, the content of the article and the conclusions have been amended and supplemented to improve the vision of the novelty of the work and its logic

Reviewer 2 Report

The manuscript entitled “Silicas with polyoxyethylene branches for modification of membranes based on microporous block copolymers.” (by Davletbaeva et al.) was completely reviewed. The manuscript investigates the designing rout of modified membranes based on microporous block copolymers. The parameters of synthesis and sorption properties were investigated.  Although some interesting results are described in the synthesis and adsorption part, the manuscript should be improved. The main concerns are :

1.       It needs more attention for English editing.

2.       The abstract should be rewritten

3.       L 44-49, 56-58, and 72-74: the references should be transferred to the end of the paragraph.

4.       There is a lot of self-citation, it should not exceed than 3 times only.

5.       Authors should support the conclusion at the last paragraph in the introduction part.

6.       The yield of the produced materials in each stage should be provided

7.       Figure 3 has a bad resolution, should be improved

8.       NMR, FTIR and TGA should have a deep discussion and support with references.  

it needs more attention for English editing

Author Response

Thank you for your comments and feedback, which have helped us to substantially improve our manuscript. We carefully investigated you recommendations and prepared comments for each item you mentioned in your review.

 The manuscript entitled “Silicas with polyoxyethylene branches for modification of membranes based on microporous block copolymers.” (by Davletbaeva et al.) was completely reviewed. The manuscript investigates the designing rout of modified membranes based on microporous block copolymers. The parameters of synthesis and sorption properties were investigated.  Although some interesting results are described in the synthesis and adsorption part, the manuscript should be improved. The main concerns are :

  1. It needs more attention for English editing.

Answer: We have done a thorough editing of the English

  1. The abstract should be rewritten

Answer: The abstract has been completely rewritten

  1. L 44-49, 56-58, and 72-74: the references should be transferred to the end of the paragraph.

Answer: The references transferred to the end of the sentence paragraph.

  1. There is a lot of self-citation, it should not exceed than 3 times only.

Answer: Only 4 self-citing references are retained, the others have been removed.

  1. Authors should support the conclusion at the last paragraph in the introduction part.

Answer: The conclusion at the last paragraph in the introduction part has been rewritten

  1. The yield of the produced materials in each stage should be provided.

Answer: The yield of products was estimated based on 29Si NMR spectroscopy data and the material balance based on the removal of the calculated amount of removed ethanol.

  1. Figure 3 has a bad resolution, should be improved

Answer: Figure 3 has been modified.

  1. NMR, FTIR and TGA should have a deep discussion and support with references.

Answer: In the discussion of the spectral characteristics of the studied compounds, an additional description was made and the corresponding references were added (Line 337-338, 388-393, 417-423).

Reviewer 3 Report

11. Some symbols such as “÷” can be replaced with more common ones.  

22. The results should be discussed in past tense.

33. Some transitions between discussions of different results, as well as explaining why a test is necessary, are recommended to make the whole manuscript more fluent, and help the reader to better follow the author.

Author Response

Thank you for your comments and feedback, which have helped us to substantially improve our manuscript. We carefully investigated you recommendations and prepared comments for each item you mentioned in your review.

The manuscript described the fabrication and characterization of polysiloxanes containing cubic and linear configurations with adjustable coordination with copper ions (ASiP-Cu). Furthermore, ASiP-Cu was incorporated in membranes based on microporous block copolymer and induced a significant membrane sorption capacity increase. Overall, the manuscript showed detailed characterization of the ASiP-Cu structure and a promising application. However, the novelty of this work needs to be stressed and some references are needed. Details are shown below:

  1. Novelty. The author stated the difficulty of preparing ASiP, compared with preparing conventional silica particles. Is this the first work describing ASiP preparation? If so, the novelty should be more stressed. If not, other works should be discussed for the audience to better understand this field. Also, what’s the advantage of ASiP? Does it show more sorption capacity improvement than conventional particles?

Answer: In the introduction, it is written that the use of sol-gel synthesis to obtain non-aggregated organosubstituted silica particles (ASiP) represents a hard task – because of the high probability of formation of cross-linked topological structures in the course of sol-gel processes (Line 84-86). In addition, it is added (Line 97-101) that works are known where the acid-catalyzed controlled hydrolytic polycondensation of TEOS provided polyethoxysiloxanes with weight-average molecular weights of 2300 –11,700, which depended on the reaction molar ratios of the water , catalyst, and solvent to TEOS. They were soluble in common organic solvents and stable to self-condensation and were characterized with high silica contents of up to 67%.

In the introductory part, it is written that we previously obtained organosubstituted silica particles ASiP-K further employed for OBC modification. Alkaline catalysis was used to obtain ASiP-K. As a result, the reaction rate was very slow, and silica had a cubic topology. In the present study, copper chloride was used as a catalyst. This led to the possibility of creating substituted silicas in both cubic and linear topology by changing the content of copper chloride in the reaction system. In addition, as the content of CuCl2 increases, it enters into coordination interaction with the polyoxyethylene branches of ASiP-Cu. In connection with the comments of the reviewer, the abstract and conclusions have been rewritten, and information on the effect of the CuCl2 content on the topology of ASiP-Cu and the properties of modified OBCs has been added.

  1. Figure resolution. Most figures (Figure 1, 2, 3, 7b, 8, 10, 14, 16, 17), especially the chemical structures are blur and a few of them are even hard to read.

Answer: Most figures have been improved.

  1. References needed.

Line 81-83: is there any previous work focusing on this ASiP topic?

Answer: The synthesis and studies of ASiP presented in this paper have not been previously carried out.

Line 300-310: is there any other work to show the absorption peaks resulted from copper coordination, and discussing the n→π* transitions?

Answer: The description has been amended (Line 337-348), the necessary reference is given.

Line 329-332: some references showing the copper-crown ether coordination and its absorption will be helpful to support the author’s statement.

Answer: Relevant references are included.

Line 385: references are needed to support “the absorption from the NMR tube glass”.

Answer: The reference was added.

  1. Line 266. “Fig. 4” should be “Fig. 3”.

Answer: Corrected.

  1. Line 278. The author stated that a further increase in CuCl2 content led to a particle size decrease. However, the particle sizes of SiP-Cu with CuCl2 contents from 0.1 to 0.5 wt% look very close to each other on Figure 3b (sample 2, 3, 4). Average size values with standard deviations may be helpful to support the statement.

Answer: Visually, the similarity of sizes is due to the fact that the scale of particle size values on the abscissa axis is non-linear. In connection with the comment, we have given a more accurate scale on the x-axis in the figure 6.

  1. Line 289-293. Photos of ASiP-Cu with different colors can be included in the supplementary information.

Answer: We have presented photos of ASiP-Cu with different colors in Figure 17.

Reviewer 4 Report

The article presents the development of microporous block copolymer membrane based on modification by silicas with polyoxyethylene branches for sorption of the organic dye. Cu concentration is adopted to control the sorption performance of the prepared membrane. The high diffusion permeability of modified OBC for molecules of organic dyes and metal ions were established. The results are interesting, useful and well discussed. There are only several minor comments before publication.

Line 435, why ASiP-Cu this high Cu (ASiP-Cu-0.1 and ASiP-Cu-0.5) concentration exhibited higher thermal resistance than that with a low Cu concentration (ASiP-Cu-0.01)?

Line 449, it is claimed that “cubic and linear polysiloxanes containing polyoxyethylene branches exhibit higher amphiphilicity compared to their coordinated linear counterparts.” How did the authors conclude this? Perhaps testing the water contact angle of OBC those modified by ASiP-Cu-X and their coordinated linear counterparts is more convincing.

It is better to explore the mechanisms of the sorption behavior. Because the Cu concentration significantly affects the structure of ASiP-Cu, which may determine the adsorption performance.

Minor:

Line 87, TEOS firstly used here should have full name.

Figure 10 is so vague that some letters are not visible.

Please simplify the conclusion

 Minor editing of English language required

Author Response

Thank you for your comments and feedback, which have helped us to substantially improve our manuscript. We carefully investigated you recommendations and prepared comments for each item you mentioned in your review.

The article presents the development of microporous block copolymer membrane based on modification by silicas with polyoxyethylene branches for sorption of the organic dye. Cu concentration is adopted to control the sorption performance of the prepared membrane. The high diffusion permeability of modified OBC for molecules of organic dyes and metal ions were established. The results are interesting, useful and well discussed. There are only several minor comments before publication.

Line 435, why ASiP-Cu this high Cu (ASiP-Cu-0.1 and ASiP-Cu-0.5) concentration exhibited higher thermal resistance than that with a low Cu concentration (ASiP-Cu-0.01)?

Answer: Clarification added to the article: Thermal resistance of ASiP-Cu-0.1 and ASiP-Cu-0.5 predominantly consisting of coordinated linear polysiloxanes with polyoxyethylene branches turned out to be higher than the heat resistance of ASiP-K and ASiP-Cu-0.01 due to significant differences in their chemical structure.

Line 449, it is claimed that “cubic and linear polysiloxanes containing polyoxyethylene branches exhibit higher amphiphilicity compared to their coordinated linear counterparts.” How did the authors conclude this? Perhaps testing the water contact angle of OBC those modified by ASiP-Cu-X and their coordinated linear counterparts is more convincing.

Answer: This conclusion was made due to the measurements of surface tension isotherms for the studied ASiP-Cu-X (Fig. 12).

It is better to explore the mechanisms of the sorption behavior. Because the Cu concentration significantly affects the structure of ASiP-Cu, which may determine the adsorption performance.

Answer: We agree with the comment.

Line 87, TEOS firstly used here should have full name.

Answer: Corrected.

Figure 10 is so vague that some letters are not visible.

Answer: Corrected.

Please simplify the conclusion

Answer: Corrected.

Round 2

Reviewer 1 Report

The work is systematic and comprehensive, but the innovation is not outstanding enough.

There are also a few grammatical problems , such as 101 lines on the third page. Language grammar issues still need to be improved.